# Concept-Centric Token Interpretation for Vector-Quantized Generative Models

**Tianze Yang** [* 1]   **Yucheng Shi** [* 1]   **Mengnan Du** [2]   **Xuansheng Wu** [1]   **Qiaoyu Tan** [3]   **Jin Sun** [1]   **Ninghao Liu** [1]

## Abstract

Vector-Quantized Generative Models (VQGMs) have emerged as powerful tools for image generation. However, the key component of VQGMs—the codebook of discrete tokens—is still not well understood, e.g., which tokens are critical to generate an image of a certain concept? This paper introduces Concept-Oriented Token Explanation (CORTEX), a novel approach for interpreting VQGMs by identifying concept-specific token combinations. Our framework employs two methods: (1) a sample-level explanation method that analyzes token importance scores in individual images, and (2) a codebook-level explanation method that explores the entire codebook to find globally relevant tokens. Experimental results demonstrate CORTEX's efficacy in providing clear explanations of token usage in the generative process, outperforming baselines across multiple pretrained VQGMs. Besides enhancing VQGMs transparency, CORTEX is useful in applications such as targeted image editing and shortcut feature detection. Our code is available at https://github.com/YangTianze009/CORTEX.

## 1. Introduction

Vector-Quantized Generative Models (VQGMs) have become powerful tools for high-quality image generation using discrete latent space representations (Ramesh et al., 2021; Esser et al., 2021; Yu et al., 2021; Jin et al., 2023; Tian et al., 2024). Despite their success, these models often exhibit concerning behaviors (e.g., showing demographic disparities in professional representation, as Fig. 1 demonstrates). Understanding these issues requires interpreting how VQGMs internally represent and process concepts. A critical com-

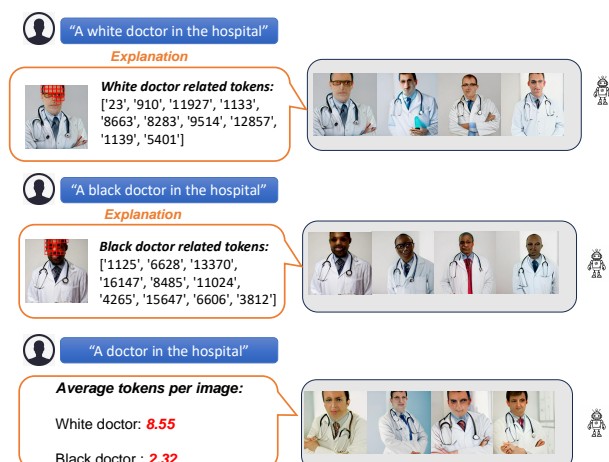

Figure 1: Token-based analysis reveals racial bias in generative models. While the model can generate both white and black doctors when explicitly prompted (top, middle), neutral prompts (bottom) show a systematic bias. CORTEX quantifies this bias through token analysis, showing white doctor-associated tokens appear nearly 4 times more frequently (8.55 vs 2.32 tokens per image).

ponent of these models is the *codebook* (Esser et al., 2021), which acts as a learned dictionary of visual elements. This codebook stores a finite set of discrete tokens, each representing various patterns or features within an image. However, not all tokens contribute equally to the generation of a particular concept (e.g., object categories or visual attributes), leading to the need for methods that can distinguish between concept-relevant and background tokens. Improving the interpretability of these token-concept relationships can help identify potential biases in the model's representations and enable precise control through targeted image editing.

Given any user-interested concept, our goal is to select tokens from the codebook whose combination can best represent it. A straightforward approach to token interpretation is to select tokens that frequently appear in images generated for a specific concept (Blei et al., 2003). However, this method often selects tokens that represent *contextual or background* elements, resulting in explanations cluttered with irrelevant information. This inability to differentiate between essential and non-essential tokens hinders a clear understanding of how the model represents concepts.

---

[*]Equal contribution  [1]School of Computing, University of Georgia [2]Department of Data Science, New Jersey Institute of Technology [3]Department of Computer Science, New York University. Correspondence to: Ninghao Liu <ninghao.liu@uga.edu>.

*Proceedings of the 42^{nd} International Conference on Machine Learning*, Vancouver, Canada. PMLR 267, 2025. Copyright 2025 by the author(s).

To address this issue, we draw on the *Information Bottleneck* principle (Tishby et al., 2000), which focuses on compressing input data while retaining the most relevant information for a given label. In the context of VQGMs, we apply this principle to train an *Information Extractor*, a module that maps image tokens back to semantic labels—reversing the information flow of generative models which typically map semantic descriptions to images.

Based on this, we propose **CORTEX** (Concept-Oriented Token Explanation) to interpret VQGMs, which utilizes the Information Extractor to understand the relation among the codebook, generated images, and visual concepts. CORTEX comprises two complementary methods: a sample-level explanation method that analyzes individual token importance in generated images, and a codebook-level explanation method that explores how codebook tokens are combined to represent specific concepts. By systematically identifying the most critical tokens and filtering out non-essential information through the Information Extractor, CORTEX provides clear, interpretable explanations of how VQGMs represent and generate specific concepts.

Experimental results demonstrate the effectiveness of CORTEX through comprehensive evaluation using various pre-trained classification models, including ResNet and Vision Transformer variants. Our sample-level explanation method reveals consistent patterns of concept-relevant tokens in generated images, while the codebook-level explanation method extends this understanding by discovering fundamental token combinations that characterize each concept. Together, these methods enhance the transparency and controllability of VQGMs, providing valuable insights into the model's internal representations and offering practical tools for downstream generative tasks such as precise image editing, as well as identifying potential biases in model representations (e.g., revealing how tokens associated with white doctors appear more frequently than those of black doctors even with a neutral prompt, as shown in Figure 1). Our main contributions are summarized as follows:

- We develop a **sample-level explanation method** that identifies concept-relevant token combinations across generated images, providing initial insights into how VQGMs represent visual concepts.

- We further introduce an **codebook-level explanation method** that extends our analysis to the entire codebook space, utilizing the same Information Extractor to discover fundamental token combinations that characterize specific concepts.

- Our experiments validate the effectiveness of our framework in enhancing VQGMs interpretability and enabling applications such as precise image editing and bias identification.

## 2. Problem Statement

### 2.1. Vector-Quantized Generative Models

Vector-Quantized Generative Models (VQGM) (Ramesh et al., 2021; Esser et al., 2021; Yu et al., 2021; Jin et al., 2023) generate images via decoding from discrete tokens. These models are typically designed for conditional generation and are capable of generating images based on given text descriptions or class labels. During the image generation process, these models typically consist of three parts: a *codebook* that stores token information, a transformer-based predictor that predicts tokens based on the codebook and the concepts, and a decoder that decodes tokens into images.

Let $G$ be a VQGM with a **codebook** $\mathcal{C} \in \mathbb{R}^{K \times d} = [t_0, \ldots, t_{K-1}]^\top$, where $K$ is the total number of unique tokens, and $t_i \in \mathbb{R}^d$ is a $d$-dimensional vector representing the token $i$. The codebook is pretrained on a large volume of image data to encode a diverse set of visual elements (Esser et al., 2021). The codebook maps continuous high-dimensional visual features into discrete tokens by serving as a look-up table. When generating an image, the model first uses its transformer predictor to predict a sequence of $m^2$ tokens according to the input concept. It then looks up each token's corresponding vector in the codebook. These vectors are arranged into a new matrix $\mathbf{E} \in \mathbb{R}^{d \times m \times m}$, which are called **token-based embeddings**. More details on how these token-based embeddings are extracted from the codebook can be found in Appendix A.1. Finally, the decoder maps this embedding $\mathbf{E}$ into an $H \times H$ image.

### 2.2. Problem Definition

Given a pretrained VQGM with a codebook $\mathcal{C} \in \mathbb{R}^{K \times d} = [t_0, \ldots, t_{K-1}]^\top$, our goal is to find token-based explanations $\mathcal{T}^* = \{\mathcal{T}_1^*, \mathcal{T}_2^*, ..., \mathcal{T}_n^*\}$ for a set of **concepts** of interest $Y = \{y_1, y_2, ..., y_n\}$, where each $\mathcal{T}_i^* \subset \mathcal{C}$ corresponds to $y_i$. Each concept $y_i \in Y$ represents a certain aspect of visual content. These concepts are specified by users, which can be contrastive (e.g., male vs. female in gender) or parallel (e.g., different object categories like "cat", "dog", and "bird"). For each concept $y_i$, its corresponding explanation $\mathcal{T}_i^*$ is a combination of tokens drawn from the codebook that characterizes how the VQGM represents the concept in its generative process: $\mathcal{T}_i^* \subseteq \{t_0, \ldots, t_{K-1}\}$.

The key challenge in explaining generative models lies in identifying truly concept-relevant tokens while filtering out background or irrelevant information. During image generation, VQGM constructs token-based embeddings $\mathbf{E} \in \mathbb{R}^{d \times m \times m}$ using high-dimensional features that are not intuitively comprehensible to humans. For instance, when explaining the concept "dog", we need to identify tokens that capture the dog's distinctive features while excluding those representing contextual elements like sky or terrain.

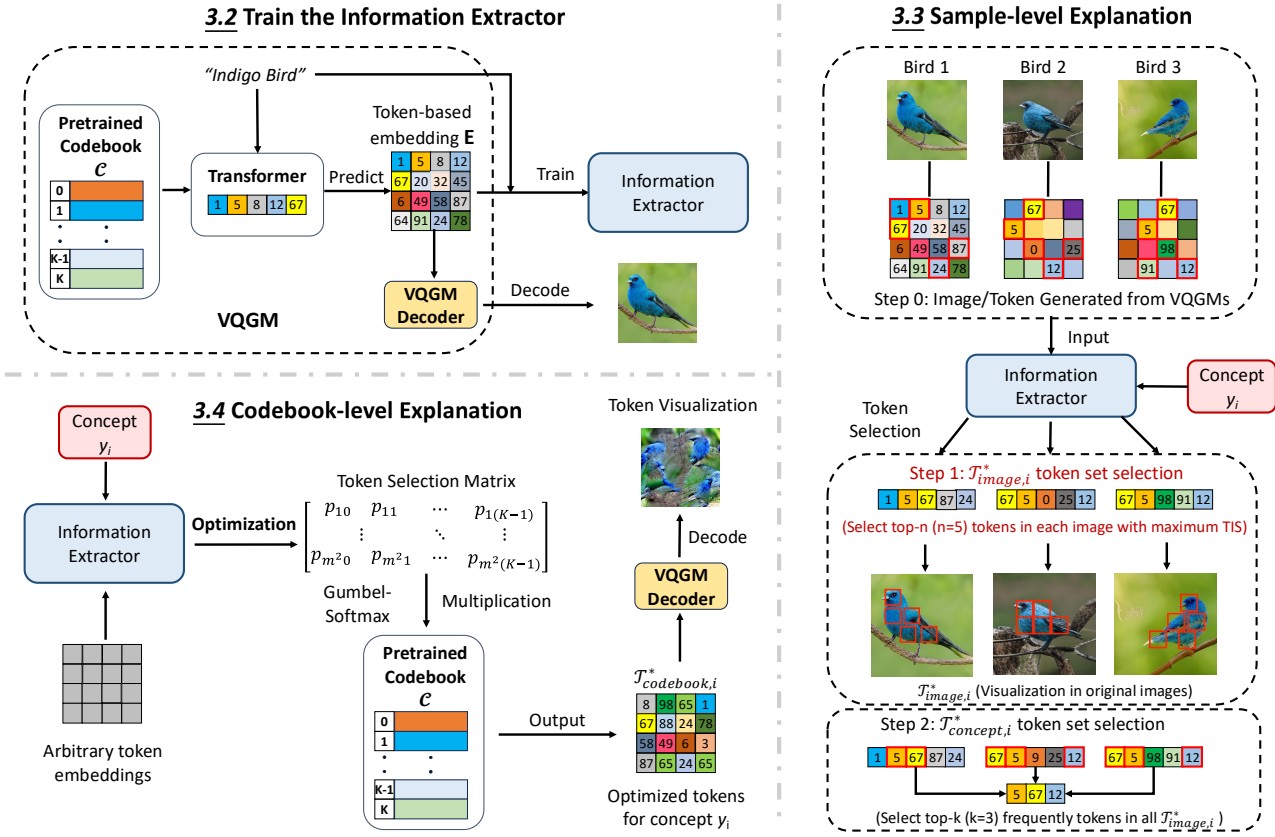

Figure 2: CORTEX has three components: an Information Extractor module and two explanation methods. The sample-level explanation method identifies concept-relevant tokens for each individual image, and the codebook-level explanation method optimizes token combinations that represent the concept.

To address this challenge of extracting concept-specific tokens, we develop an information extractor in Section 3.2.

# 3. Methodology

## 3.1. Overview

Our framework of concept-specific token explanation for VQGMs is shown in Figure 2. We start by developing an **information extractor** which maps codebook tokens to concepts. Based on the information extractor, we develop two complementary methods to explain how VQGMs generate images with codebook tokens. The first method, called **sample-level explanation**, reveals how tokens form concepts in actual generated images. The second method, called **codebook-level explanation**, interprets how VQGMs represent concepts in codebook space. These explanations illuminate how users can identify potential weaknesses in VQGMs and make targeted edits to generated images.

## 3.2. Information Extractor

The Information Extractor Module (IEM) serves as the foundation of our framework. Unlike generative models that

produce images based on textual descriptions and concepts, the IEM works in reverse, mapping from images (codebook tokens) to concepts. Given a token-based embedding $\mathbf{E} \in \mathbb{R}^{d \times m \times m}$ of an image, the trained IEM $f$ generates a probability distribution over all concepts:

$$\mathbf{p} = f(\mathbf{E}) \in [0,1]^n, \tag{1}$$

where $\mathbf{p} = [p_1, p_2, ..., p_n]$ is a probability distribution over the concepts $Y = \{y_1, y_2, ..., y_n\}$, with each $p_i$ representing the probability that concept $y_i$ is present in the embedding $\mathbf{E}$. This formulation allows us to quantify how strongly each concept is represented in any given token-based embedding.

To train the IEM, we first use VQGM to generate a training dataset by using each concept $y_i \in Y$ as either text prompts or class conditions to generate images. For each image, we also obtain its corresponding token-based embedding $\mathbf{E}$. The IEM is trained as a supervised classifier to map embeddings to their respective concepts, learning to recognize concept-specific token patterns within VQGM's output.

The design of our IEM inherently aligns with the Information Bottleneck (IB) principle (Tishby et al., 2000). To accurately predict the probabilities for each concept, the

IEM must *identify and extract the most informative features while discarding irrelevant ones*. This natural requirement for discriminative feature extraction mirrors the core idea of the IB principle, which seeks to compress input information while preserving task-relevant features. Previous studies have shown that such compression leads to more interpretable representations (Bang et al., 2021; Yang et al., 2022), as it forces the model to focus on the essential features of each concept. By leveraging this property, we can extract explanations from the IEM by contrastively analyzing how different tokens represent the given concepts. Given the trained IEM $f$ and concepts $Y$, we aim to find the optimal token combinations $\mathcal{T}^* = \{\mathcal{T}_1^*, \mathcal{T}_2^*, ..., \mathcal{T}_n^*\}$ that explain each concept:

$$\mathcal{T}^* = \phi(f, [\mathbf{E}], Y), \tag{2}$$

where $\phi$ represents either our sample-level explanation method that requires input embeddings $\mathbf{E}$ to analyze token importance in specific images, or our codebook-level explanation method that explores the entire codebook space without needing the Embedding $\mathbf{E}$.

### 3.3. Sample-level Explanation

We first propose a sample-level explanation method based on token importance analysis. Formally, given a token-based embedding $\mathbf{E} \in \mathbb{R}^{d \times m \times m}$ generated by the VQGM, we compute the saliency score $S_i$ (Simonyan, 2013; Chefer et al., 2021; Selvaraju et al., 2020; Smilkov et al., 2017) of each token with respect to each concept $y_i$ as:

$$S_i = \frac{1}{N} \sum_{l=1}^{N} \nabla_{\mathbf{E}} f_{y_i}(\mathbf{E} + \boldsymbol{\epsilon}_l), \tag{3}$$

where $f_{y_i}(\mathbf{E})$ represents the prediction probability of concept $y_i$ (i.e., $p_i$), $N$ is the number of samples, and $\boldsymbol{\epsilon}_l \sim \mathcal{N}(0, \sigma^2 \mathbf{I})$ with $\sigma = \alpha(\max(\mathbf{E}) - \min(\mathbf{E}))$. The resulting $S_i \in \mathbb{R}^{d \times m \times m}$ has the same dimensions as $\mathbf{E}$.

We then calculate the Token Importance Score (TIS) for each token $t_j$ in the $m \times m$ grid with respect to each concept $y_i$. The TIS serves as a measure of the importance or relevance of the token to the prediction of concept $y_i$, with higher values indicating higher importance. $\text{TIS}(t_j, y_i)$ is computed by taking the maximum value across all $d$ channels of the gradient at the specific position corresponding to the token $t_j$ as follows:

$$\text{TIS}(t_j, y_i) = \max_{1 \leq k \leq d} |S_i(k, p_j)|, \tag{4}$$

where $p_j$ represents the position of token $t_j$ in the $m \times m$ grid, and $S_i(k, p_j)$ denotes the saliency score at position $p_j$ in the $k$-th channel for concept $y_i$. This operation reduces the $d$-dimensional gradient vector at each token's position to a scalar, representing the token's relevance to concept $y_i$.

After calculating the TISs, we identify relevant token combinations for each concept $y_i$:

$$\mathcal{T}_{\text{image},i}^* = \text{Top-}n(t_j : j \in 1, \ldots, m^2, \text{key} = \text{TIS}(\cdot, y_i)),$$
$$\mathcal{T}_{\text{concept},i}^* = \text{Top-}k(\bigcup_{\substack{\text{sampled images}}} \mathcal{T}_{\text{image},i}^*, \text{key} = \text{Freq}).$$

$$\tag{5}$$

Here, $\mathcal{T}_{\text{image},i}^*$ represents the Top-$n$ tokens with the highest TIS for each specific image with respect to concept $y_i$, providing an image-specific explanation. It identifies the most distinguishable tokens for generating the target image. $\mathcal{T}_{\text{concept},i}^*$ aggregates these image-specific sets across all sampled images related to concept $y_i$ and selects the $k$ most frequent tokens, offering a concept-specific explanation.

### 3.4. Codebook-level Explanation

While sample-level explanation reveals how tokens are utilized in specific generated images, it only examines existing token patterns in generated data. To better understand the semantic meaning of model components within VQGMs, we propose an optimization-based method that directly explores the **entire codebook space**. For any concept $y_i \in Y$, this approach directly searches for token combinations that best characterize the concept.

Given concept $y_i$, we aim to find a **token selection matrix** $\mathbf{P} \in \mathbb{R}^{m^2 \times K}$, where $m^2$ is the total number of token positions in the embedding $\mathbf{E}$, and $K$ is the size of the codebook. Each row of $\mathbf{P}$ contains a probability distribution over the $K$ possible tokens. We employ a binary mask $\mathbf{M}_{\text{mask}}$ to specify the image regions where the tokens are optimized.

Since the direct selection of discrete tokens from the codebook is not differentiable, we employ Gumbel-Softmax (Jang et al., 2016) for differentiable token selection. This technique transforms the discrete selection process into a continuous, differentiable operation, enabling the use of gradient-based optimization algorithms to find the optimal token combinations.

$$\mathbf{E} = \text{GumbelSoftmax}(\mathbf{P}, \tau) \times \mathcal{C}, \tag{6}$$

where $\text{GumbelSoftmax}(P, \tau) \in \{0, 1\}^{m^2 \times K}$ is a one-hot matrix representing the selected tokens, $\tau$ is the temperature parameter, and $\mathcal{C} \in \mathbb{R}^{K \times d}$ is the codebook matrix (details in Appendix A.4). The optimization of $\mathbf{P}$ is conducted by:

$$\mathbf{P_{k+1}} = \mathbf{P_k} - \eta(\nabla_P \mathcal{L}(\mathbf{P_k}) \odot \mathbf{M}_{\text{mask}}),$$
$$\mathcal{L}(\mathbf{P}) = -f_{y_i}(\mathbf{E}) + \alpha \|\mathbf{E}\|_2^2, \tag{7}$$

where $\alpha$ is a regularization parameter and $\eta$ is the learning rate. After optimization converges, we obtain the final selection matrix $\mathbf{P}^*$. The optimal token combination $\mathcal{T}_i^*$ for

concept $y_i$ is derived from $\mathbf{P}^*$ for the unmasked positions:

$$\mathcal{T}^*_{codebook,i} = \{t_k : k = \arg\max_j \mathbf{P}^*_{\mathbf{l,j}}, \forall l \text{ where } \mathbf{M}_{\text{mask},\mathbf{l}} = 1\}. \tag{8}$$

By applying this process to each concept in $Y$, we can obtain the complete set of token-based explanation $\mathcal{T}^*_{codebook} = \{\mathcal{T}^*_{codebook,1}, \mathcal{T}^*_{codebook,2}, ..., \mathcal{T}^*_{codebook,n}\}$ that shows how VQGMs represent different concepts at the codebook level.

Together, these complementary approaches provide both sample and global perspectives on how VQGMs encode conceptual information in their token-based representations.

### 3.5. Applications of Token-based Explanations

Our framework's token-level explanations offer several practical applications:

- **Shortcut Feature Detection.** Using our sample-level explanation $\mathcal{T}^*_{\text{concept}}$, we can quantitatively detect if short-cut features or concepts are involved in image generation. For instance, given demographic concepts $y_1$ and $y_2$ (e.g., "white doctor" and "black doctor"), we can measure bias by comparing the frequencies of their corresponding token sets $\mathcal{T}^*_{\text{concept},1}$ and $\mathcal{T}^*_{\text{concept},2}$ in images generated from neutral prompts (e.g., "doctor"). The systematic differences in these frequencies provide a statistical measure of the model's inherent biases.

- **Targeted Image Editing.** Our codebook-level explanation method enables precise concept manipulation by optimizing token selection matrix $\mathbf{P}$ within a specified mask region $\mathbf{M}_{\text{mask}}$. Given a target concept $y_i$, we optimize the tokens in the masked region to maximize $f_{y_i}(\mathbf{E})$, effectively steering the image representation towards concept $y_i$ while preserving tokens outside $\mathbf{M}_{\text{mask}}$. The optimized token set $\mathcal{T}^*_{\text{codebook},i}$ then provides a controlled way to edit local image regions according to the target concept while maintaining the integrity of surrounding content.

## 4. Experiments

Our proposed framework aims to explain concept-specific information in VQGMs on a diverse range of concepts. The experiments are designed to verify that the token combinations $\mathcal{T}^*$ selected by CORTEX are indeed the most relevant to the concepts $Y$ being explained.

### 4.1. Experimental Setup

To validate our proposed explanation method, we adopt the methodology of treating each category in the ImageNet challenge dataset as a distinct concept $y_i$ to be explained (Chefer et al., 2021; Binder et al., 2016; Simonyan, 2013; Deng et al., 2009). To validate that our selected tokens are indeed the

Table 1: Comparison of IEMs prediction accuracy (%).

| Model | Top-1 | Top-3 | Top-5 | Top-10 |
|---|---|---|---|---|
| CNN-based Extractor | 53.07 | 71.37 | 77.73 | 84.65 |
| ResNet-based Extractor | 51.43 | 69.23 | 76.00 | 83.12 |
| Transformer-based Extractor | 48.71 | 66.74 | 73.63 | 81.43 |

most crucial for concept representation, we evaluate our method through a counterfactual approach: if these tokens are truly critical, masking them should significantly impact the model's ability to recognize the target concept. To ensure the robustness of our evaluation, we employ four well-established pretrained classification models as benchmarks. The selected pretrained models include variants of ResNet (He et al., 2016) (ResNet18 and ResNet50) and Vision Transformer (ViT) (Dosovitskiy, 2020) (ViT-B/16 and ViT-B/32). These models represent state-of-the-art approaches in image recognition. To elucidate the selected $\mathcal{T}^*$ from our proposed explanation method $\phi$ of VQGMs, we use a synthetic data set generated by VQGAN (Esser et al., 2021), which encompasses the same categories as ImageNet (synthetics dataset details in Appendix A.2).

In our experiment, the IEM $f$ is trained as an image classifier with $1,000$ ImageNet categories. We train 3 IEMs with different architectures: (1) CNN-based Extractor (CE), (2) ResNet-based Extractor (RE), and (3) Transformer-based Extractor (TE). IEMs' architectures can be found in the Appendix A.3. All IEMs take token-based embedding $\mathbf{E} \in \mathbb{R}^{256 \times 16 \times 16}$ as input and output probability distributions over 1000 ImageNet labels. Based on Table 1, the performance of our models, with Top-1 accuracies of $53.07\%$, $51.43\%$, and $48.71\%$ for CNN-based, ResNet-based, and Transformer-based architectures, respectively, is significant considering the potential inaccuracies introduced by VQGAN image generation process. Despite this challenge, the high Top-5 accuracies (exceeding $73\%$) demonstrate that our IEMs effectively capture the relationship between token patterns and image labels.

### 4.2. Sample-level Explanation Evaluation

**Experimental Design.** In this part, our experiments aim to validate that the token combinations $\mathcal{T}^*_{\text{image}}$ and $\mathcal{T}^*_{\text{concept}}$ identified by our sample-level explanation method are indeed the most relevant and representative of specific concepts. In our image-specific analysis, given each concept $y_i$ in $Y$, we define $\mathcal{T}^*_{\text{image},i}$ as the set of the Top-$n$ tokens with the highest TIS in images containing concept $y_i$, where $n$ ranges from 5 to 50 in increments of 5. For each token in $\mathcal{T}^*_{\text{image},i}$, we mask the corresponding regions in these images and measure the change in softmax probability for concept $y_i$ across four pretrained models: ViT-B/16, ViT-B/32, ResNet18, and ResNet50. As a baseline, we randomly select $n$ tokens and mask the associated regions in the same images to compare

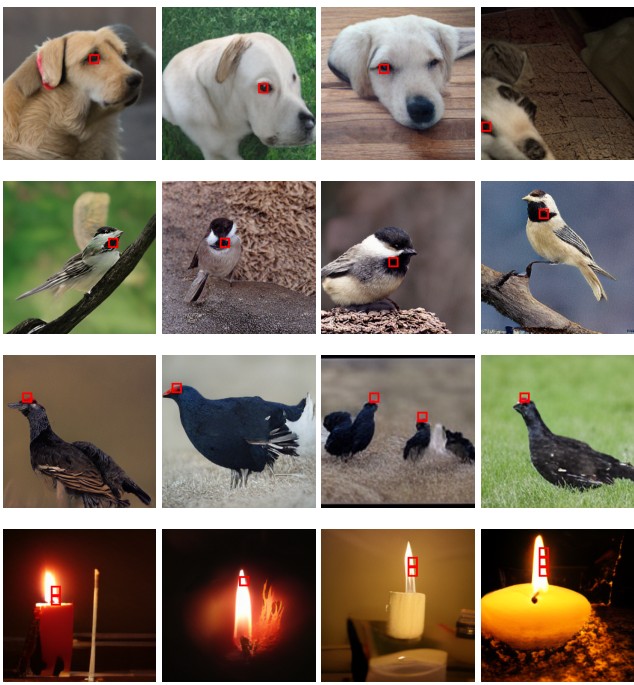

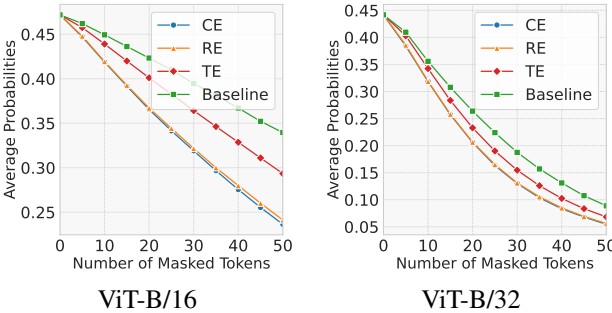

ViT-B/16          ViT-B/32

Figure 4: Image-specific sample-level explanation method evaluation results: average probabilities vs. number of masked tokens.

Figure 3: Token visualization for different concepts using our sample-level explanation method. Each row shows 4 images of a distinct concept. Red boxes highlight high-TIS tokens, revealing consistent identification of class-specific features (e.g., eyes, neck, red crowns, flame) across images.

the effect with masking tokens in $\mathcal{T}^*_{\text{image},i}$.

In our concept-specific analysis, given each concept $y_i$, we first identify the Top-$n$ ($n = 20$) highest-TIS tokens in each training image containing concept $y_i$ to form individual $\mathcal{T}^*_{\text{image},i}$ sets. From the union of these sets across all training images containing the concept $y_i$, we select the Top-$k$ ($k = 100$) most frequent tokens to form $\mathcal{T}^*_{\text{concept},i}$. We compare this with a frequency-based baseline of selecting the Top-100 most frequent tokens in all images containing the concept $y_i$ without using our IEM $f$. We then mask the corresponding patches of selected tokens in test images containing the concept $y_i$ and measure the change in the probability of the target concept. Our analyses use the 1000 ImageNet classes as our set of concepts of interest $Y = y_1, y_2, ..., y_{1000}$, allowing us to evaluate our method's effectiveness across a diverse range of visual concepts.

**Image-specific Evaluation Results.** Figure 3 demonstrates the efficacy of our sample-level explanation method in identifying concept-related features across multiple images (more results can be found in Appendix A.7). Each row in the figure represents a distinct label. For each label, we present 4 different images. Within each image, we highlight a specific token in $\mathcal{T}^*_{\text{image},i}$ using a red bounding box. These visualizations demonstrate our sample-level explanation method's ability to focus on tokens that often

correspond to specific, concrete visual features within each concept. For instance, the consistent highlighting of eyes, red crowns, and other distinctive features across multiple images of the same class indicate that these tokens can effectively represent meaningful, class-specific characteristics.

Quantitatively, Figure 4 shows the average change in softmax probability for specific labels as we mask from 5 to 50 high-TIS tokens. Across two SOTA pretrained ViT models, our method consistently leads to a steeper decline in probability compared to random selection, demonstrating its effectiveness in identifying tokens crucial to concept representation. Notably, both CNN-based and ResNet-based IEMs exhibit similar declining trends, suggesting that different models attend to similar tokens for specific concepts, while the Transformer-based IEM shows relatively lower performance. This lower performance of the Transformer-based IEM can be attributed to its weaker classification capabilities, which may result in less accurate information learning and token importance estimation. These results validate our sample-level explanation method's ability to identify label-relevant features and provide interpretable insights into VQGMs.

**Concept-specific Evaluation Results.** Table 2 presents the concept-specific sample-level evaluation results across 4 different pretrained models. When masking tokens based on our $\mathcal{T}^*_{\text{concept},i}$, we mask fewer tokens on average ($n_1 = 42.176$, $n_2 = 40.629$, and $n_3 = 45.552$ for the CNN-based, ResNet-based, and Transformer-based extractors, respectively) compared to the baseline ($n_b = 64.166$). Despite masking smaller regions, our CNN-based and ResNet-based methods achieve superior performance—for instance, with ResNet50, they decrease accuracy by 46.6% and 46.8% and probability by 0.403 and 0.404, respectively, compared to baseline's 41.9% and 0.365. While our Transformer-based extractor shows comparable performance to the baseline (e.g., 42.4% vs 42.2% accuracy drop for ResNer18), it also achieves this with fewer masked tokens. These significant changes in prediction accuracy with fewer masked tokens indicate that our $\mathcal{T}^*_{\text{concept}}$ successfully identifies regions con-

Table 2: Concept-specific sample-level explanation evaluation results (Acc: prediction accuracy, P: probability, $n$: number of masked tokens, $\Delta A$: change in accuracy, $\Delta P$: change in probability).

| Pretrained Model | Acc | P | CE | | | RE | | | TE | | | Baseline | | |
|---|---|---|---|---|---|---|---|---|---|---|---|---|---|---|
| | | | $n$ | $\Delta A \downarrow$ | $\Delta P \downarrow$ | $n$ | $\Delta A \downarrow$ | $\Delta P \downarrow$ | $n$ | $\Delta A \downarrow$ | $\Delta P \downarrow$ | $n$ | $\Delta A \downarrow$ | $\Delta P \downarrow$ |
| **ResNet18** | 55.6% | 0.452 | | -45.4% | -0.381 | | **-45.5%** | **-0.382** | | -42.4% | -0.358 | | -42.2% | -0.356 |
| **ResNet50** | 56.1% | 0.472 | 42.176 | -46.6% | -0.403 | 40.629 | **-46.8%** | **-0.404** | 45.552 | -43.1% | -0.374 | 64.166 | -41.9% | -0.365 |
| **ViT-B/16** | 59.0% | 0.472 | | -9.50% | -0.112 | | **-9.60%** | **-0.113** | | -7.01% | -0.089 | | -7.30% | -0.090 |
| **ViT-B/32** | 58.0% | 0.442 | | **-36.0%** | **-0.289** | | **-36.0%** | **-0.289** | | -32.7% | -0.260 | | -33.2% | -0.264 |

taining crucial information for concept $y_i$. This demonstrates that our IEMs effectively fulfill their role in the Information Bottleneck framework—identifying and preserving the most essential concept-specific tokens while filtering out irrelevant ones, regardless of the architecture used.

### 4.3. Discussion

Our counterfactual masking experiments demonstrate that removing the tokens identified by CORTEX substantially degrades pretrained classification models' performance, indicating that these tokens indeed capture the most discriminative features for each target concept. By contrast, randomly masking the same number of tokens leads to relatively smaller performance drops. This comparative analysis confirms the effectiveness of our sample-level explanation method in pinpointing concept-critical tokens, thereby validating the interpretability and reliability of CORTEX's token selections.

### 5. Application of Explanations

This section demonstrates practical applications of our CORTEX framework through two key use cases. First, we show how CORTEX can be used to detect and quantify biases in text-to-image models by analyzing concept-specific token distributions. Second, we illustrate CORTEX's capability in targeted image editing by optimizing specific concept-relevant tokens.

### 5.1. Bias Detection in Text-to-Image Models

To demonstrate CORTEX's capability in detecting biases within a specific text-to-image model, Dalle-mini (Dayma et al., 2021), we conducted experiments focusing on two common types of biases: gender and color representation in professional settings. We specifically examined the case of doctor representations, as professional representation bias has been a significant concern in generative models.

**Experimental Setup.** Our experiment consisted of two phases. In the first phase, we generated training data using specific prompts to establish baseline representations.

Table 3: IEM Classification Accuracy

| Bias Type | Accuracy (%) |
|---|---|
| Gender | 99.7 |
| Color | 99.95 |

For color bias analysis, we used the prompts "A black doctor in the hospital" and "A white doctor in the hospital". For gender bias analysis, similar prompts were used to generate images representing male and female doctors. We generated 10,000 images for each prompt, dividing them into training (8,000), validation (1,000), and test (1,000) sets. These images were used to train our IEM as a binary classifier, achieving high classification accuracy, as shown in Table 3. Using the trained IEMs, we applied our sample-level explanation method to identify tokens that characterize each concept (color & gender). This provided us with a set of tokens that could serve as indicators of potential biases in the model's representations.

Table 4: Comparison of Mean Frequency (per image) and Cliff's $\delta$ for different attributes.

| | Mean Frequency (per image) | | | Cliff's $\delta$ | | |
|---|---|---|---|---|---|---|
| | **Top-5** | **Top-10** | **Top-20** | **Top-5** | **Top-10** | **Top-20** |
| **Color** | | | | | | |
| White | 4.80 | 6.61 | 8.55 | 0.311 | 0.418 | **0.456** |
| Black | 2.35 | 2.18 | 2.32 | | | |
| **Gender** | | | | | | |
| Male | 7.37 | 8.27 | 5.65 | 0.359 | **0.394** | 0.264 |
| Female | 3.27 | 3.43 | 3.45 | | | |

**Bias analysis with the Neutral Prompt.** In the second phase, we generated $2,000$ images using the neutral prompt "A doctor in the hospital" to investigate potential biases. For each demographic attribute, we analyzed the frequency of tokens from their concept-specific token sets ($\mathcal{T}^*_{\text{concept},male}$, $\mathcal{T}^*_{\text{concept},female}$, $\mathcal{T}^*_{\text{concept},white}$, and $\mathcal{T}^*_{\text{concept},black}$). Following Equation 5, we identify the Top-$n$ tokens ($n = 5, 10, 20$) with highest TIS in each training image, then select Top-$k$ ($k = 10$) most frequent tokens to form each concept-specific set. As shown in Table 4, using Cliff's $\delta$ (detailed in Appendix A.6), tokens from $\mathcal{T}^*_{\text{concept},white}$ appear nearly four times more frequently than $\mathcal{T}^*_{\text{concept},black}$ (8.55 vs 2.32 tokens per image when $n = 20$, $\delta = 0.456$). For gender, tokens from $\mathcal{T}^*_{\text{concept},male}$ appear more than twice as fre-

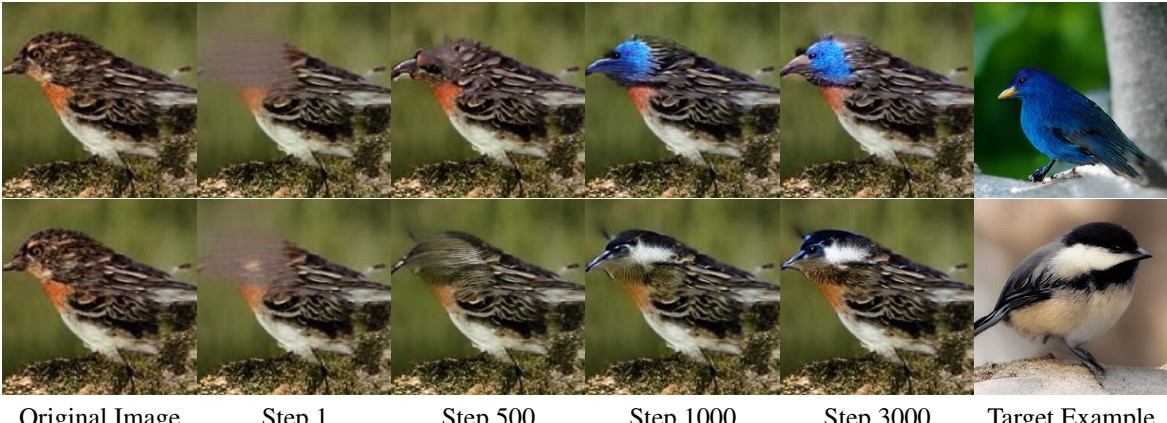

| Original Image | Step 1 | Step 500 | Step 1000 | Step 3000 | Target Example |

Figure 5: Codebook-level optimization-based image editing process.

quently as $\mathcal{T}^*_{\text{concept},female}$ (8.27 vs 3.43 tokens per image when $n = 10, \delta = 0.394$). This systematic bias in token frequencies directly corresponds to observable patterns in the model's outputs, when given the neutral prompt "a doctor in the hospital", the model consistently generates images of white male doctors. These results demonstrate COR-TEX's effectiveness in detecting and quantifying biases in text-to-image models.

## 5.2. Image Editing

Figure 5 shows the visualization of the image editing process using our token selection codebook-level explanation method proposed in section 3.4. By applying $\mathbf{M}_{\text{mask}}$ to constrain optimization to only the tokens corresponding to the bird's head region, these sequences demonstrate the gradual transformation of one bird species into another while preserving other regions. Using the target bird species as concept $y_i$, our method optimizes $\mathcal{T}^*_{\text{codebook},i}$ within the masked region, resulting in progressive changes to features like beak shape and color. This controlled transformation illustrates our method's ability to identify and manipulate concept-specific tokens effectively.

## 5.3. Discussion

Our sample-level explanation detects shortcut features or biases by analyzing how frequently certain concept tokens appear under neutral prompts. Meanwhile, the codebook-level explanation enables localized image editing by selectively refining tokens in a chosen region, providing both interpretability and precise control in generative modeling.

## 6. Related Work

**Vector quantization in computer vision.** Vector quantization has been instrumental in advancing image generative models (Gray, 1984; Nasrabadi & Feng, 1988). VQ-VAE (Van Den Oord et al., 2017) pioneered the use of discrete latent codes for efficient image reconstruction, overcoming "posterior collapse" issues in VAEs. DALL-E (Ramesh et al., 2021) extended this to text-to-image generation, while VQGAN (Esser et al., 2021) and ViT-VQGAN (Yu et al., 2021) enhanced image quality through perceptual and adversarial objectives. In video generation, MAGVIT (Yu et al., 2023), VideoPoet (Kondratyuk et al., 2023), and LaVIT (Jin et al., 2024; 2023) applied vector quantization for spatial-temporal modeling and multimodal learning. FSQ (Mentzer et al., 2023) simplifies vector quantization with efficient scalar quantization, while MAGE (Li et al., 2023) combines masked token modeling with generative objectives to improve image representation learning. Our work builds upon these VQGMs, offering a novel approach to interpreting discrete tokens and providing insights into visual information encoding and utilization.

**Vision model explainability.** Traditional approaches to explaining vision models primarily fall into two categories: heatmap-based methods (Sundararajan et al., 2017; Selvaraju et al., 2020; Binder et al., 2016; Gandelsman et al., 2023; Chefer et al., 2021; Wang et al., 2020; 2024; Boggust et al., 2023; Jetley et al., 2016; Gupta et al., 2022; Du et al., 2018), which highlight influential image regions, and optimization-based methods (Nguyen et al., 2016b; Erhan et al., 2009; Yosinski et al., 2015; Nguyen et al., 2015; Simonyan, 2013; Nguyen et al., 2016a; FEL et al., 2023; Goh et al., 2021; Zimmermann et al., 2021), which generate synthetic inputs to maximize specific activations. While insightful, these pixel-level approaches are limited in explaining VQGMs. Our CORTEX approach extends this to the token level, providing concept-specific explanations of discrete latent representations. This offers deeper insights into VQGMs' generative processes, bridging traditional and modern explainability.

# 7. Conclusion

In this paper, we introduced CORTEX, a framework that interprets VQGMs through concept-specific token analysis guided by the Information Bottleneck principle. Our experiments show that CORTEX effectively identifies tokens critical to concept representation, enabling not only fine-grained image editing but also systematic bias detection by examining token distributions for sensitive attributes. By revealing how codebook tokens capture high-level concepts, CORTEX enhances transparency and controllability in generative models. Future work includes extending CORTEX to a broader range of VQGMs and vision-language models (Shi et al., 2025), such as video-based models, and further exploring its potential usability (Wu et al., 2024) in model improvement.

# Acknowledgements

The work is, in part, supported by NSF (#IIS-2223768). The views and conclusions in this paper are those of the authors and should not be interpreted as representing any funding agencies.

# Impact Statement

Our research has significant implications for understanding and detecting biases in AI systems, particularly in generative models. Our experimental results revealed concerning demographic disparities in image generation, specifically showing systematic biases in professional representation where tokens associated with certain demographics appeared significantly more frequently than others in neutral prompts. These findings highlight the importance of developing robust methods for detecting and quantifying such biases in AI systems.

While our work contributes to model interpretability and bias detection, we acknowledge that identifying biases is only the first step toward addressing broader issues of fairness and representation in AI systems. We encourage future work to build upon these findings to develop more equitable generative models, while remaining mindful of both the capabilities and limitations of interpretability tools in addressing societal challenges.

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

# A. Appendix

## A.1. Codebook Extraction in VQGMs

Vector-Quantized Generative Models (VQGMs) utilize a codebook as a learned dictionary of discrete tokens to represent high-dimensional visual features. During the encoding process, input images are passed through an encoder to produce continuous latent representations, which are then quantized by mapping each vector to its nearest neighbor in the codebook using vector quantization. The resulting sequence of discrete token indices forms the compressed representation of the image.

Formally, let $\mathbf{z} \in \mathbb{R}^d$ be a latent vector produced by the encoder, and $\mathcal{C} = \{\mathbf{t}_0, \mathbf{t}_1, \ldots, \mathbf{t}_{K-1}\}$ be the codebook with $K$ codebook entries, where each $\mathbf{t}_i \in \mathbb{R}^d$. The quantized output is obtained via:

$$\mathbf{t}^* = \arg\min_{\mathbf{t}_i \in \mathcal{C}} \|\mathbf{z} - \mathbf{t}_i\|_2^2$$

This process is repeated for each location in the latent grid (e.g., $16 \times 16$), resulting in a grid of token indices that compactly represent the input image. These tokens are then used as input to the decoder for image reconstruction or generation.

## A.2. Symthetic Dataset

To evaluate our proposed methods, we utilize a synthetic dataset generated by VQGAN (Esser et al., 2021). Using a synthetic dataset rather than real-world images offers two key advantages: (1) it provides direct access to the token-based embeddings $\mathbf{E}$ during the generation process, enabling us to analyze how the model encodes concepts at the token level, and (2) it ensures that the data perfectly matches the token distribution learned by the generative model, eliminating potential distribution gaps between training and evaluation. This synthetic dataset encompasses all ImageNet categories, allowing us to directly examine how the generative model utilizes tokens to encode concept-specific information.

The dataset consists of:

- 1,000,000 training images
- 300,000 validation images
- 50,000 test images

The images are evenly distributed across all ImageNet categories, resulting in 1,000, 300, and 50 images per category in the training, validation, and test sets, respectively. Each generated image has a resolution of $256 \times 256$ pixels and is represented by a $16 \times 16$ grid of tokens. During the generation process, we obtain both the generated image and its corresponding token-based embedding $\mathbf{E}$, enabling direct analysis of the relationship between tokens and visual concepts.

## A.3. Information Extractor

This appendix provides details on the structure of two Information Extractor: CNN-based model and Resnet-based model.

### A.3.1. CE: CNN-BASED EXTRACTOR

The CNN-based Extractor (CE) is a convolutional neural network designed for image classification. The model comprises two main blocks, each containing four convolutional layers (conv1_1 to conv1_4 and conv2_1 to conv2_4). Each convolutional layer utilizes 512 filters with a $3 \times 3$ kernel size, stride of 1, and padding of 1, followed by batch normalization and ReLU activation. Max pooling ($2 \times 2$ kernel, stride 2) is applied after each block. The network concludes with three fully connected layers: the first transforms $512 \times 4 \times 4$ input features to 4096 output features, the second maintains 4096 features, and the final layer maps to the number of classes. Additional features include batch normalization and ReLU activation after the first two fully connected layers, with dropout (0.5) applied after the first fully connected layer.

### A.3.2. RE: RESNET-BASED EXTRACTOR

The ResNet-based Extractor (RE) is an advanced model incorporating residual connections and Squeeze-and-Excitation (SE) blocks. The network consists of two main layers, each containing 3 residual blocks. Each residual block comprises two

convolutional layers ($3 \times 3$ kernel, maintaining channel size) with batch normalization and ReLU activation, a shortcut connection, and an SE block for channel-wise attention. The SE block employs global average pooling followed by two fully connected layers with reduction and expansion, using sigmoid activation for generating attention weights. The model concludes with global average pooling to reduce spatial dimensions, followed by two fully connected layers: 512 to 2048 features, and 2048 to the number of classes. Batch normalization and ReLU activation are applied after the first fully connected layer, with dropout (0.5) implemented.

### A.3.3. TE: Transformer-based Extractor

The Transformer-based Extractor (TE) processes input features $[B, 256, 16, 16]$ by reshaping them into 256 sequences of $16 \times 16$ patches. These patches are projected to 512 dimensions through a linear layer with LayerNorm and GELU activation. A learnable class token is prepended, and learnable positional embeddings are added after scaling by $\sqrt{512}$. The backbone comprises 12 encoder layers with Pre-LN and dual-stream normalization. Each encoder contains 8-head self-attention with dropout (0.2) and a feed-forward network ($512 \rightarrow 2048 \rightarrow 512$) with GELU activation. Both attention and feed-forward outputs undergo BatchNorm before residual connections. For classification, the class token passes through LayerNorm and a two-layer MLP ($512 \rightarrow 1024$ with LayerNorm, GELU, and dropout), followed by projection to class logits.

Both CE, RE, and TE are designed to process input tensors with 256 channels and spatial dimensions of $16 \times 16$ tokens. While the Transformer-based architecture demonstrates marginally lower performance metrics, this outcome aligns with our deliberate design choice to maintain a lightweight model structure. Given IEM's requirement for computational efficiency, we implemented a relatively shallow Transformer architecture, which may constrain its potential to surpass CNN-based models in this specific token pattern recognition task. Nevertheless, the overall performance demonstrates both architectures' capability to effectively learn meaningful representations from token-based input embeddings $\mathbf{E}$.

### A.3.4. Training Settings

**Training setting for CE and RE.** These information extractors were trained using a batch size of 256 for 80 epochs, with the task involving classification across 1000 classes. We employed the Adam optimizer with an initial learning rate of $0.001$ and weight decay of $1e - 4$. To adjust the learning rate during training, we implemented a StepLR scheduler, which decreased the learning rate by a factor of $0.1$ every 20 epochs. The loss function used for training was Cross Entropy Loss. Our experimental setup allowed for the training of multiple model architectures under consistent conditions, enabling fair comparison of their performance.

**Training setting for TE.** Due to the distinct characteristics of transformer-based architectures, we adopted a specialized training strategy different from CNN and ResNet approaches. Specifically, we employed AdamW optimizer with weight decay 1e-4, combined with a hybrid learning rate schedule consisting of a linear warmup phase (10% of total iterations) followed by cosine annealing. The initial learning rate was set to 1e-3. For training stability and efficiency, we implemented mixed-precision training using automatic mixed precision (AMP) with gradient scaling. The model was trained for 80 epochs with a batch size of 256. This configuration addresses the optimization challenges specific to transformer architectures while maintaining stable gradient flow throughout the training process.

## A.4. Gumbel-Softmax Technique for Token Selection Optimization

In our implementation, we employ the Gumbel-Softmax technique to optimize the selection of tokens from the codebook. This method enables differentiable sampling from a discrete distribution, which is essential for our gradient-based optimization process. The core of our approach involves a matrix P of shape $(256, 16384)$, where each row represents a probability distribution over the codebook tokens.

The Gumbel-Softmax approximation operates by adding Gumbel noise to the logits (log probabilities) derived from P at each optimization step. The Gumbel-Max trick states that for a categorical distribution with class probabilities $p_i$, sampling can be performed as:

$$\text{argmax}_i(\log(p_i) + g_i) \tag{9}$$

where $g_i$ are i.i.d. samples from Gumbel(0, 1) distribution.

We then apply a softmax function with a temperature parameter $\tau$ to these noisy logits:

$$y_i = \frac{\exp((\log(p_i) + g_i)/\tau)}{\sum_j \exp((\log(p_j) + g_j)/\tau)} \tag{10}$$

In the "hard" version of this technique, we convert this soft distribution to a one-hot vector by selecting the maximum value:

$$y_{\text{hard}} = \text{onehot}(\text{argmax}_i(y_i)) \tag{11}$$

The final output is then:

$$y = \text{stop\_gradient}(y_{\text{hard}} - y) + y \tag{12}$$

This process allows us to sample discrete tokens while maintaining differentiability, thereby enabling backpropagation through the sampling process.

A key feature of the Gumbel-Softmax is the temperature parameter $\tau$, which controls the discreteness of the samples. As $\tau$ approaches zero, the samples become more discrete, closely approximating one-hot vectors. Conversely, as $\tau$ increases, the distribution becomes smoother and more uniform.

Throughout the optimization process, we update the P matrix based on the gradients computed through this differentiable sampling procedure. The gradient of the Gumbel-Softmax estimator with respect to the logits is:

$$\frac{\partial y_i}{\partial \log(p_k)} = \frac{y_i(\delta_{ik} - y_k)}{\tau} \tag{13}$$

where $\delta_{ik}$ is the Kronecker delta.

By utilizing this approach, we can optimize the selection of discrete tokens from the codebook in a manner compatible with gradient-based optimization methods. This compatibility is crucial for our objective of maximizing the activation of target labels in our classification model.

The Gumbel-Softmax technique thus serves as a bridge between the discrete nature of our token selection problem and the continuous optimization landscape required for effective gradient-based learning. It allows us to backpropagate through the discrete sampling operation, enabling end-to-end training of our model while maintaining the ability to produce discrete outputs during inference.

### A.5. Codebook-level Explanation Evaluation

**Experimental Design.** To validate that our codebook-level explanation method can effectively identify concept-specific token combinations $\mathcal{T}^*_{\text{codebook}}$, we provide comprehensive quantitative experimental results. Due to the computational intensity of multi-step optimization in codebook-level explanation, we conduct our evaluation using CNN-based (CE) and ResNet-based (RE) IEMs, as the Transformer-based IEM (TE) has significantly more parameters and would require prohibitively long optimization times. We conduct experiments using 10 bird categories (500 images in total) from the synthetic dataset. These bird images can be generated by VQGAN in high quality, and the 4 pretrained models achieve high prediction accuracies on these 500 images, ranging from $84.6\%$ to $90.2\%$ (refer to $\text{Acc}_{\text{Orig}}$ in Table 5).

We pair 10 bird categories into 5 groups, each category serving as both original and target labels. For each category, we utilize 50 images from the test set. In our experimental design, when images from a category serve as the original images, its paired category becomes the target label, and vice versa. For instance, in the pair (Goldfinch, Water Ouzel), when Goldfinch images are being optimized, Water Ouzel serves as the target label, and conversely, when Water Ouzel images are used as original images, Goldfinch becomes the target label. This reciprocal design is applied consistently across all five pairs depicted in Figure 6. Every image in our dataset undergoes optimization as an original image, with its paired category serving as the target label. In this experiment, each concept $y_i$ is a specific bird category.

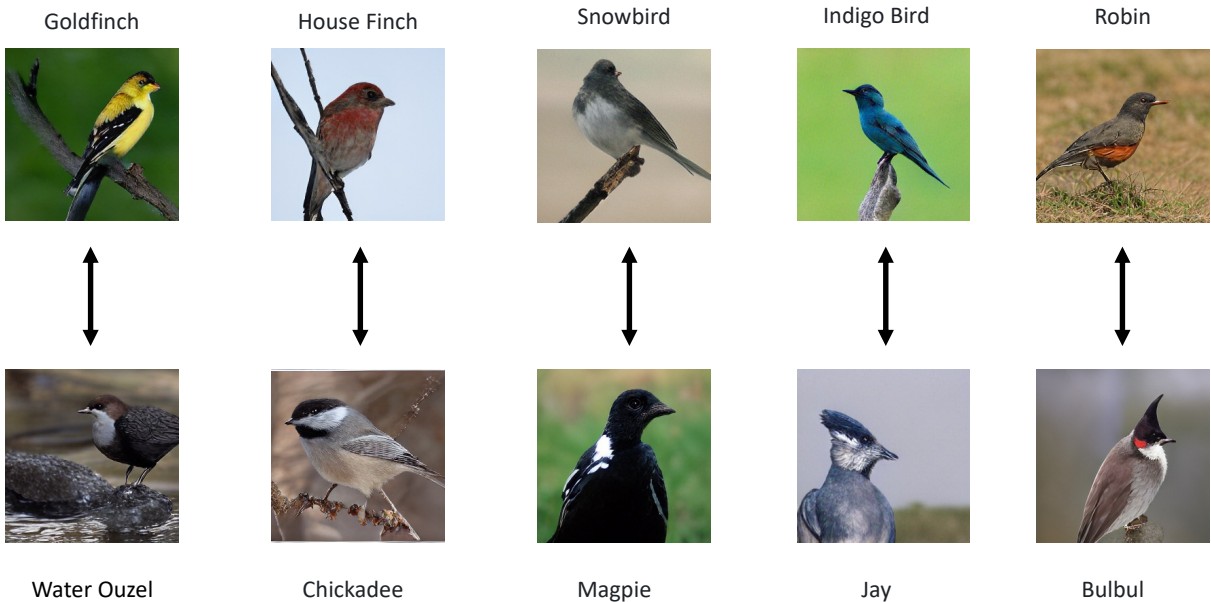

Figure 6: codebook-level explanation method experiment design

The optimization process begins with token-based embeddings $\mathbf{E}$ from the original label images, optimizing towards the target label, which is the concept to be explained. We focus on a fixed $4 \times 4$ region within the $16 \times 16$ token grid, limiting $\mathcal{T}^*_{\text{codebook},i}$ to 16 tokens (only $1/16$ of total 256 tokens). We evaluate the changes in softmax probabilities for both the original and target labels across four pretrained models.

We explore two optimization methods, both aiming to maximize the activation of a target bird label (the concept to be explained): 1) *Token selection optimization (our method)*: We optimize a token selection matrix, which represents the probability of selecting each token from the codebook for specific positions in the target region. 2) *Embedding optimization (baseline)*: We directly optimize the $d$-dimensional embedding in the target region. After optimization, we apply vector quantization (Gray, 1984) to map each optimized embedding vector to the nearest token in the codebook, minimizing the Euclidean distance. Optimized embeddings were decoded via VQGAN to generate images. To measure the effectiveness of each method optimization method, we calculate the change in probabilities for the original and target labels before and after optimization:

$$\Delta P_{\text{Orig}} = P_{\text{Orig}}(\text{optimized}) - P_{\text{Orig}}(\text{initial}),$$
$$\Delta P_{\text{Targ}} = P_{\text{Targ}}(\text{optimized}) - P_{\text{Targ}}(\text{initial}). \tag{14}$$

**Evaluation Results.** Table 5 shows the results of our optimization methods across different models. The *Token Selection* method consistently outperforms the *Embedding Optimization* baseline by both reducing the original label probability ($\Delta P_{\text{Orig}}$) and increasing the target label probability ($\Delta P_{\text{Targ}}$). For instance, in the ResNet18 model with CNN-based extractor, our method decreases the probability of the original label by $36.5\%$ (from 0.795 to 0.505) and increases the probability of the target label about 11 times (from 0.016 to 0.178) compared to the initial probability. In contrast, the Embedding Optimization baseline achieves a $28.2\%$ decrease in the original label probability and a 7.9-fold increase in the target label probability. This shows that our method surpasses the baseline by achieving a greater reduction in the original label and a more significant increase in the probability of the target label. Similar improvements are observed in another information extractor. In the ViT-B/16 model with ResNet-based extractor, our Token Selection method reduces the probability of the original label by $31.7\%$ and increases the probability of the target label by over 40 times, significantly outperforming the baseline. These results indicate that our Token Selection method effectively identifies the most important tokens contributing to the target label. By directly optimizing the token selection matrix end-to-end, it finds the token combination that

maximally activates our information extractor $f$. In contrast, the Embedding Optimization method optimizes embeddings and then maps them back to the nearest tokens in the codebook, which may result in suboptimal token combinations due to the lack of end-to-end optimization.

These results validate the effectiveness of our codebook-level explanation method in identifying and manipulating class-relevant features within VQGMs. The substantial increases in target label probabilities, often by more than an order of magnitude, demonstrate the method's potential for enhancing model interpretability and its applicability in targeted image editing tasks

### A.6. Cliff's $\delta$

Cliff's $\delta$ is a non-parametric effect size measure that quantifies the degree of overlap between two groups of observations. For two groups X and Y with sizes $N_x$ and $N_y$, it is calculated as:

$$\delta = \frac{\sum_{i=1}^{N_x} \sum_{j=1}^{N_y} [I(x_i > y_j) + 0.5 I(x_i = y_j)] - \frac{N_x N_y}{2}}{N_x N_y} \tag{15}$$

where $I(\cdot)$ is the indicator function. The value ranges from -1 to 1, with larger absolute values indicating stronger effect sizes. Following Romano et al. (Romano & Kromrey, 2006), the effect size can be interpreted as: negligible for $|\delta| < 0.147$, small for $|\delta| < 0.33$, medium for $|\delta| < 0.474$, and large otherwise. In our analysis, X and Y represent the occurrence frequencies of specific concepts ($\mathcal{T}^*$) across different images. A positive $\delta$ indicates a significant difference in the activation frequencies between two groups, suggesting potential bias in concept representation.

### A.7. More Visualizatoin Results

Figure 7 presents additional sample-level explanation visualization results, complementing the analysis provided in Section 4.2. The figure is structured into two distinct sets of four rows each, each set focusing on a specific token for a particular category. This approach demonstrates the efficacy of our sample-level explanation method in identifying concept-specific features across multiple images. The first four rows showcase visualizations related to the "black grouse" category, highlighting a single, consistently meaningful token across different images of this bird species. Similarly, the subsequent four rows are dedicated to visualizations of the "candle" category, emphasizing the same token across various candle images. In each image, we highlight the token exhibiting a high TIS using a red bounding box. These visualizations illustrate our sample-level explanation method's ability to focus on tokens that frequently correspond to specific, concrete visual features within each concept. For instance, the consistent highlighting of particular features (such as the red crown for the black grouse or flame for candles) across multiple images of the same class indicates that these tokens effectively represent meaningful, class-specific characteristics. By consistently focusing on the same token within each category, we demonstrate our method's ability to extract and emphasize stable, category-relevant features across diverse visual representations.

Table 5: codebook-level explanation method evaluation results.

| Model | Acc$_{Orig}$ | P$_{Orig}$ | P$_{Targ}$ | $\Delta$P$_{Orig}$ $\downarrow$ / $\Delta$P$_{Targ}$ $\uparrow$ | | | |
|---|---|---|---|---|---|---|---|
| | | | | **Embedding Optimization** | | **Token Selection** | |
| | | | | **CE** | **RE** | **CE** | **RE** |
| **ResNet18** | 86.8% | 0.795 | 0.016 | -0.224 / 0.127 | -0.202 / 0.097 | **-0.290 / 0.162** | -0.284 / 0.160 |
| **ResNet50** | 84.6% | 0.780 | 0.011 | -0.217 / -0.122 | -0.206 / 0.102 | **-0.282 / 0.165** | -0.277 / 0.155 |
| **ViT-B/16** | 89.0% | 0.766 | 0.003 | -0.193 / 0.095 | -0.174 / 0.072 | -0.237 / **0.128** | **-0.243** / 0.121 |
| **ViT-B/32** | 90.2% | 0.758 | 0.003 | -0.196 / 0.090 | -0.190 / 0.080 | -0.239 / 0.112 | **-0.245 / 0.120** |

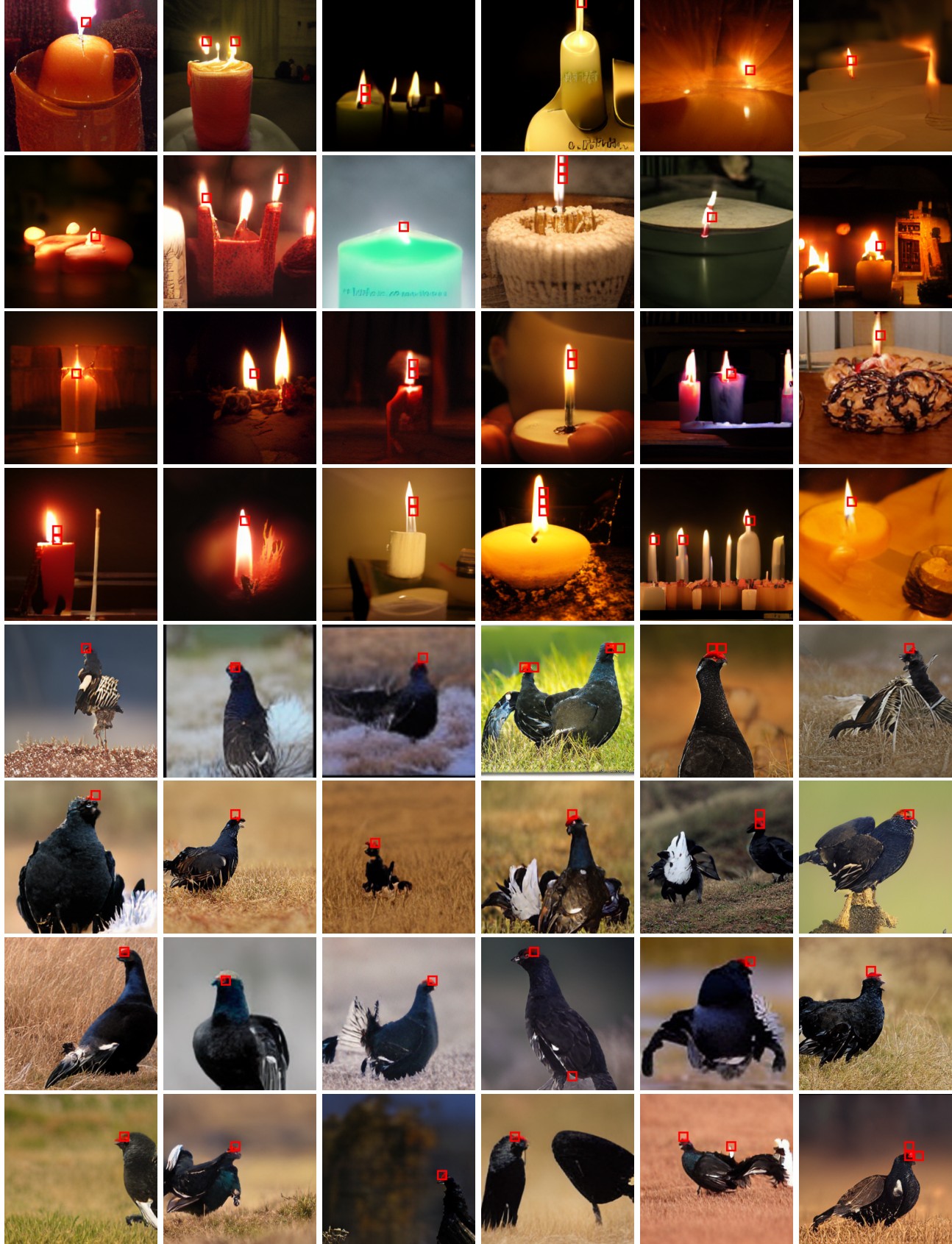

Figure 7: Token visualization in different categories: red box in each row represents the same token

