# OpenReview forum: "Concept-Centric Token Interpretation for Vector-Quantized Generative Models"
_ICML.cc/2025/Conference — ICML 2025 poster_

### Official Review · Reviewer_YRHi · 2025-03-11

**Overall Recommendation:** 3

**Summary:**

This paper introduces CORTEX, an approach for interpreting Vector-Quantized Generative Models (VQGMs) by identifying concept-specific token combinations from their codebooks. The authors develop two complementary methods: a sample-level explanation that analyzes token importance in individual images and a codebook-level explanation that searches the entire codebook for globally relevant tokens representing specific concepts. Experimental evaluations show CORTEX's ability to provide clear explanations of token usage in the generative process while also enabling applications such as targeted image editing and bias detection.

## update after rebuttal

The authors have addressed my main questions and concerns. For me, this is a self-consistent and complete work. I have also carefully read and am aware of the other reviewers' concerns. Overall, I maintain my original borderline accept score, but I would not be surprised if it were rejected.

**Claims And Evidence:**

Yes

**Essential References Not Discussed:**

None

**Experimental Designs Or Analyses:**

The sample-level explanation evaluation compares against random token selection, which is a weak baseline. A fairer comparison would include other attribution methods adapted to the token space.

**Methods And Evaluation Criteria:**

Yes

**Other Comments Or Suggestions:**

- Instead of using top-n and top-k token selection in the sample-level explanation, the authors could consider using a threshold-based approach that better adapts to the importance distribution.

**Other Strengths And Weaknesses:**

## pros

- Addresses an interesting interpretability gap in VQGMs.
- Visualizations illustrate the concept-token relationships identified by the method.
- The paper is well written and easy to follow.

## cons

- See the questions below.

**Questions For Authors:**

- The sample-level explanation uses gradient-based attribution, which has known issues like gradient saturation and noisy gradients. How does this affect the reliability of token importance scores?
- About the codebook-level explanation, what guarantees the process consistently converges to meaningful token combinations rather than adversarial patterns that simply maximize class probability?
- The main evaluation relies on masking tokens and measuring probability changes, but this assumes tokens have independent effects. How does the method account for the interdependencies between tokens where combinations matter more than individual tokens?
- How does CORTEX handle tokens that might be relevant to multiple concepts simultaneously? The framework assumes a clear concept-to-token mapping that may not exist in practice.
- The term "concept" may not be suitable. The proposed method primarily focuses on entities or objects. How well does CORTEX generalize to more abstract concepts like "happy" or "dangerous" that don't have clear visual correspondences? If not suitable for these abstract concepts, I suggest using "object-centric" instead of "concept-centric".

**Relation To Broader Scientific Literature:**

This work is related to vector quantization, information bottleneck principles, and bias detection in generative models.

**Theoretical Claims:**

There are no theoretical claims.

---

> ### Author Rebuttal · Authors · 2025-04-01
>
> Thank you for your constructive feedback. We address your concerns:
>
> ### Weak baseline comparison:
> Our comparison against both random selection and the frequency-based baseline (Table 1) highlights CORTEX’s effectiveness in identifying concept-relevant tokens. While the frequency-based method offers a stronger baseline than random selection, our method achieves a greater reduction in concept probability using fewer masked tokens. This suggests that CORTEX can better filter out contextual information, whereas frequency-based selection tends to pick background tokens that are not strongly tied to the concept itself.
>
> ### Threshold-based approach:
> Thank you for the suggestion. While threshold-based selection can serve a similar role as the top-n or top-k strategy by selecting the most important tokens based on their importance scores, it requires careful tuning for each VQGM and IEM. In contrast, our top-n and top-k strategies provide a consistent and architecture-independent evaluation across models.
>
> ### Gradient-based attribution issues:
> To reduce the impact of noisy gradients, we adopt the SmoothGrad [1] principle by adding noise to the input embeddings and averaging the resulting gradients, thus producing more stable token attributions. Furthermore, our sample-level explanation $\mathcal{T}^*_{\text{concept}}$ also aggregates highly activated tokens across multiple images via Equation 5, effectively mitigating the issue of gradient saturation that may occur in individual samples.
>
> ### Convergence guarantees:
> We acknowledge that this issue may arise with certain concepts, but we have quantitatively analyzed the performance of codebook-level explanations in Table 5 of our appendix. This analysis demonstrates that convergence to adversarial patterns is not the dominant case.
>
> ### Token interdependencies:
> Our codebook-level explanation can explain the interdependencies between tokens because this method optimizes tokens within a region simultaneously, obtaining the token combination that best represents this concept. We don't optimize tokens one by one but rather optimize the token selection matrix, and the optimization process considers the interdependencies between tokens.
>
> ### Multi-concept tokens:
> Tokens can indeed be relevant to multiple concepts. The token importance scores (TIS) are concept-specific, allowing us to map the same token to different concepts with varying importance levels. Although a single token may be associated with multiple concepts, it can represent different concepts when combined with different other tokens.
>
> ### "Concept-centric" terminology:
> Our method can explain abstract concepts, not just entities or objects. For example, in Section 5, we demonstrate how CORTEX explains relatively abstract concepts like "male" and "female" which go beyond simple visual objects. Furthermore, in text-to-image generative models like Dalle, our approach can explain any concept by identifying the most relevant token combinations when concepts such as "happy" or "dangerous" are used as prompts to generate images. We thank the reviewer for this thoughtful suggestion and will consider using more precise terminology in the final version of our paper.
>
> [1] Smilkov, Daniel, et al. "Smoothgrad: removing noise by adding noise." ICML 2017.

---

> > ### Comment · Reviewer_YRHi · 2025-04-08
> >
> > I thank the authors for the response, which has addressed most of my concerns. For now, I will keep my score, and I will also follow the authors' discussion with other reviewers.

---

> > > ### Author Response · Authors · 2025-04-08
> > >
> > > Thank you for your response. We're glad that we could address your concerns.

---

### Official Review · Reviewer_4bpB · 2025-03-13

**Overall Recommendation:** 3

**Summary:**

This paper introduces CORTEX, a method for interpreting Vector-Quantized Generative Models (VQGMs) with concept-oriented token explanations. CORTEX employs sample-level and codebook-level explanations. CORTEX is useful for shortcut detection (i.e. biases) and image editing. For the evaluation, they CORTEX employs a synthetic dataset generated by VQGAN with ImageNet categories. Experiments show that CORTEX enhances transparency and controllability in generative models.

**Claims And Evidence:**

Yes

**Essential References Not Discussed:**

No

**Experimental Designs Or Analyses:**

Yes, the VQGMs setting is legit.

**Methods And Evaluation Criteria:**

Yes, however the experimental setting is limited (see weaknesses below).

**Other Comments Or Suggestions:**

No

**Other Strengths And Weaknesses:**

### Strengths

- The proposed approach identifies the most important token used by VQGMs, enhancing their transparency.

- The paper is relevant to the community, especially for shortcut detection.

### Weaknesses

- The evaluation setting is limited: only a VQGM is used (GAN-based), concepts are limited to ImageNet classes.

- Lack of comparisons to other methods. Results with different kinds of information extraction networks are not sufficient. The authors should explain why their method, for example, better identifies biases than others.

- Some aspects in the proposed methodology are unclear. The figures and notation for example should be improved. See questions below.

- More details on how the extraction of codebook works in VQGM should be explained better; an additional section in the Appendix could benefit the paper's readability.

**Questions For Authors:**

- What are the implications of tokens that activate less frequently? Have the underrepresented concepts generated images, for example, lower quality?

- In Figure 3, which are the token-based embeddings? The coloured squares? The figures need additional notation used in the paper.

- Given a class, i.e. "Indigo Bird", are there multiple possible token-based embeddings $E$?

- Equation 3 is unclear. What is changing in the summation with different $l$s? $E$? Maybe an index notation on $E$ would be clearer. Why add noise $\epsilon_l$?

- Equation 4, why max? What about taking the average of the channels instead?

- The letter 'k' is used multiple times in different contexts (codebook, top-k etc.). A different letter for different things is recommended to improve readability.

- Are there more meaningful metrics other than frequency or cliff to identify biases?

- Does the concept label space $Y$ need to be the same as the label-conditioned space in the VQGM?

**Relation To Broader Scientific Literature:**

Neural networks are very well known black box models affected by different biases. This work contributes to interpretability methods for VQGMs.

**Theoretical Claims:**

This is not a theoretical paper, however, it relies on the Information Bottleneck principle in compression. They use an Information Extractor network to filter out background and non-essential tokens. The visualizations in the paper seem to support this principle.

---

> ### Author Rebuttal · Authors · 2025-04-01
>
> Thank you for your detailed review. We address your concerns and questions below:
>
> **Limited evaluation setting:**
> Our evaluation is not limited to VQGAN or ImageNet classes. We also include a text-to-image VQGM, DALL·E in section 5.1, where concepts are defined by prompts such as “male/female/black/white doctor”. These experiments demonstrate that CORTEX can explain arbitrary, user-defined concepts beyond fixed class labels. We also include additional results on the SOTA VQGM, VAR [1]. Evaluation using 10,000 VAR-generated images confirms that CORTEX effectively identifies concept-critical tokens.
>
> | Pretrained Model | Top-10 (ours) | Top-10 (random) | Top-20 (ours) | Top-20 (random) | Top-30 (ours) | Top-30 (random) |
> | ---------------- | ------------- | --------------- | ------------- | --------------- | ------------- | --------------- |
> | ViT-B/32         | **3.9**       | 1.3             | **9.4**       | 3.2             | **15.8**      | 6.7             |
> | ResNet50         | **11.8**      | 4.5             | **31.3**      | 22.5            | **49.4**      | 46.5            |
>
> *Table: Prediction probability drop after masking tokens from CORTEX on VAR-generated images; Larger drops indicate that the masked tokens are more important.*
>
> **Lack of comparisons:**
> As the first work to explain VQGMs in their token space, we have limited baseline options for comparison. However, we also established non-trivial baselines except for the random ones, including the frequency-based baseline in Table 2 and the embedding optimization baseline in Table 5. We evaluate CORTEX with different IEM architectures and find it consistently identifies concept-related tokens, demonstrating architecture-independent interpretability.
>
> **Unclear methodology aspects:**
> We will improve figures and notation in the revision. Specifically:
>
> 1. **Less frequently activated tokens:**
>    Less frequently activated tokens represent elements that contribute minimally to a specific concept. These typically correspond to **background information** like sky or grass that lack the distinctive characteristics of the concept.
>
> 2. **Token-based embeddings:**
>    In Figure 3, the colored grids represent the token-based embedding. Each position contains a token from the codebook. We will include the notation for **E** in Figure 2 in the revision.
>
> 3. **Multiple embeddings per class:**
>    Yes, there can be multiple possible token-based embeddings for a class like "Indigo Bird." This is why we train our IEM to identify the token combination that **best** represents the concept.
>
> 4. **Equation 3 clarification:**
>    Equation (3) is inspired by SmoothGrad [2], which adds noise to the input multiple times and averages the resulting gradients to reduce the model’s sensitivity to small perturbations, yielding a more stable saliency score.
>
> 5. **Max in Equation 4:**
>    Since each channel captures different aspects of feature importance, we use the maximum across channels to identify the most discriminative feature at each position, following the max-selection approach from Section 3.1 of [3]. To validate this, we compare max and average strategies by measuring the drop in prediction probability after masking top-ranked tokens. The max operation consistently causes greater drops, indicating it better captures concept-relevant tokens.
>
> | Pretrained Model | Top-10 (max) | Top-10 (average) | Top-20 (max) | Top-20 (average) | Top-30 (max) | Top-30 (average) |
> | ---------------- | ------------ | ---------------- | ------------ | ---------------- | ------------ | ---------------- |
> | ViT-B/32         | **12.3**     | 11.7             | **23.6**     | 22.7             | **31.2**     | 29.8             |
> | ResNet50         | **25.3**     | 23.7             | **36.3**     | 35.1             | **41.4**     | 40.5             |
>
> *Table: Performance comparison across average and max operation on VQGAN-generated images.*
>
> 6. **Letter 'k' is used multiple times:**
>    Thank you for your suggestion. We will improve our notations in the revision.
>
> 7. **Different metrics for bias:**
>    Cliff's δ provides a standardized measurement that accounts for distribution differences. Other metrics like Jensen-Shannon divergence could be used, but Cliff's δ offers clear interpretability with established thresholds for effect size (small/medium/large).
>
> 8. **Concept label space:**
>    Yes, they are the same: the concept labels we want to explain are exactly the same as the labels (or text prompts) used to condition the VQGM.
>
> **Extraction of codebook:**
> We will add a section in the Appendix to provide more details.
>
> [1] Tian, Jiang, et al. "Visual autoregressive modeling: Scalable image generation via next-scale prediction." NeurIPS, 2024.
> [2] Smilkov, et al. "Smoothgrad: removing noise by adding noise." ICML, 2017.
> [3] Simonyan, et al. "Deep inside convolutional networks: Visualising image classification models and saliency maps." arXiv preprint.

---

### Official Review · Reviewer_EKNf · 2025-03-13

**Overall Recommendation:** 4

**Summary:**

The paper introduces Concept-Oriented Token Explanation (CORTEX), a framework for interpreting Vector-Quantized Generative Models (VQGMs). CORTEX employs sample-level and codebook-level explanation methods to identify concept-specific tokens, enhancing the transparency of how VQGMs generate images. By using an Information Extractor Model based on the Information Bottleneck principle, CORTEX effectively distinguishes critical tokens from background elements. Experimental results demonstrate its superiority over baseline methods in explaining token usage, with practical applications in detecting biases and enabling targeted image editing. CORTEX thus provides valuable insights into the internal workings of VQGMs, paving the way for more interpretable and controllable generative models.

**Claims And Evidence:**

Claim 1: CORTEX Enhances VQGM Interpretability

Evidence: Comprehensive evaluations show CORTEX outperforms baseline methods in explaining token usage, revealing consistent patterns of concept-relevant tokens in generated images.

Claim 2: CORTEX Identifies Concept-Specific Tokens

Evidence: The Information Extractor Model (IEM) effectively captures the relationship between token patterns and image labels, with high Top-5 accuracies demonstrating accurate identification of relevant tokens.

Claim 3: CORTEX Detects Biases in Generative Models

Evidence: Experiments with neutral prompts reveal systematic biases, with tokens associated with certain demographics appearing more frequently, quantified using statistical measures like Cliff’s $\delta$.

Claim 4: CORTEX Enables Targeted Image Editing

Evidence: Visualizations and quantitative comparisons show CORTEX can precisely manipulate image content by optimizing concept-relevant tokens, leading to significant changes in target label probabilities.

**Essential References Not Discussed:**

All key references have been cited.

**Experimental Designs Or Analyses:**

This work identifies the Top-n (n = 20) highest-TIS tokens and Top-k (k = 100) most frequent tokens. How do the parameters n and k affect the performance?
This work is only based on the data generated by VQGAN. What is the effect of the data generated by the latest SOTA model such as MAGE and FSQ?

**Methods And Evaluation Criteria:**

The methods used in the paper include the development of the Concept-Oriented Token Explanation (CORTEX) framework, which consists of two main approaches: sample-level explanation and codebook-level explanation. The sample-level explanation method analyzes token importance scores in individual images to identify concept-specific tokens, while the codebook-level explanation method explores the entire codebook to find globally relevant tokens using an Information Extractor Model (IEM) based on the Information Bottleneck principle. The evaluation criteria involve comprehensive experimental validation using various pretrained classification models to assess the effectiveness of CORTEX in providing clear explanations of token usage, detecting biases by analyzing token frequencies in neutral prompts, and enabling precise image editing by optimizing concept-relevant tokens. The performance is measured through changes in softmax probabilities for original and target labels, comparison of token importance scores, and statistical measures like Cliff’s $\delta$ to quantify biases.

**Other Comments Or Suggestions:**

No other comments.

**Other Strengths And Weaknesses:**

CORTEX provides a comprehensive framework that includes both sample-level and codebook-level explanations. This dual approach allows for detailed analysis at both the individual image level and the broader codebook level, offering a more complete understanding of how VQGMs generate images.

**Questions For Authors:**

No other questions.

**Relation To Broader Scientific Literature:**

CORTEX extends these explainability techniques to the token level, offering concept-specific explanations for discrete latent representations in VQGMs. This provides a deeper understanding of the generative processes in these models, bridging the gap between traditional pixel-level explanations and modern token-based approaches.

**Theoretical Claims:**

The CORTEX framework significantly enhances the interpretability of Vector-Quantized Generative Models (VQGMs) by providing detailed, concept-specific explanations of how these models represent and generate images. By identifying and analyzing the importance of discrete tokens, CORTEX offers insights into the internal mechanisms of VQGMs, making them more transparent and understandable.

---

> ### Author Rebuttal · Authors · 2025-04-01
>
> Thank you for your positive assessment of our work. We address your questions below:
>
> ### Parameters n and k:
> **n=20** represents the top tokens with the highest Token Importance Scores (TIS) selected from each image, while **k=100** represents the most frequently occurring tokens across all sample images. Both parameters face similar trade-offs:
>
> If the values are too small, we might miss tokens that significantly contribute to the concept or fail to capture the full diversity of concept representations.
> On the other hand, if the values are too large, we risk including irrelevant tokens that dilute the concept representation or include more background information.
>
> Figure 4 in the main paper demonstrates how performance changes as n varies from 5 to 50, showing that as the number of masked tokens increases, prediction probability continuously decreases. However, the rate of this decreasing gradually diminishes, indicating that the initial few tokens contain the core information about the concept, while including more tokens begins to incorporate less relevant information.
>
> ### Latest SOTA models
> Our proposed **CORTEX** can be applied to any type of vector-quantized generative model.
> Following your suggestions, we further validate CORTEX on the latest SOTA VQGM, **VAR [1]** published in NeurIPS 2024. Specifically, we trained an Information Extractor Model (IEM) using token-based embeddings generated by VAR. We mask the Top-10, Top-20, and Top-30 tokens selected by CORTEX and compare the drop in prediction probability of pretrained ViT and ResNet against randomly selected tokens. Evaluation using 10,000 VAR-generated images confirms that CORTEX effectively identifies concept-critical tokens.
>
> | Pretrained Model | Top-10 (ours) | Top-10 (random) | Top-20 (ours) | Top-20 (random) | Top-30 (ours) | Top-30 (random) |
> | ---------------- | ------------- | --------------- | ------------- | --------------- | ------------- | --------------- |
> | ViT-B/32         | **3.9**       | 1.3             | **9.4**       | 3.2             | **15.8**      | 6.7             |
> | ResNet50         | **11.8**      | 4.5             | **31.3**      | 22.5            | **49.4**      | 46.5            |
>
> **Table:** Prediction probability drop after masking tokens from CORTEX on VAR-generated images; Larger drops indicate that the masked tokens are more important.
>
> We will also include a discussion of models such as **MAGE** and **FSQ** in the revised version.
>
> [1] Tian, Jiang, et al. "Visual autoregressive modeling: Scalable image generation via next-scale prediction." *NeurIPS*, 2024.

---

### Official Review · Reviewer_1ugt · 2025-03-15

**Overall Recommendation:** 3

**Summary:**

* This paper presents CORTEX (Concept-Oriented Token Explanation), a novel framework for interpreting Vector-Quantized Generative Models (VQGMs). VQGMs have become powerful in image generation, but the role of their codebook tokens in representing concepts remains unclear.


* The authors identify the problem of differentiating concept - relevant tokens from background ones in VQGMs. To address this, they draw on the Information Bottleneck principle to develop an Information Extractor. This extractor maps codebook tokens to concepts, calculating Token Importance Scores (TIS) to find optimal token combinations for each concept.
CORTEX consists of two complementary methods. The sample - level explanation method analyzes token importance scores in individual images. It computes saliency scores for tokens, uses Gumbel - Softmax for differentiable token selection, and optimizes a token selection matrix to identify concept - relevant tokens in generated images. The codebook - level explanation method explores the entire codebook space. It directly searches for token combinations that best represent specific concepts, optimizing a token selection matrix within a specified mask region.


* The authors conduct extensive experiments using various pretrained classification models and a synthetic dataset generated by VQGAN. They train three Information Extractor Models (IEMs) with different architectures and evaluate the performance of CORTEX through counterfactual evaluation. Results show that CORTEX effectively identifies tokens critical to concept representation. The sample - level explanation method can highlight concept - related features in images, and the codebook - level explanation method outperforms the baseline in identifying and manipulating class - relevant features.


* The framework has practical applications. It can be used for shortcut feature detection, such as detecting biases in text - to - image models by analyzing concept - specific token distributions.

**Claims And Evidence:**

The claims made in the submission are clear

**Essential References Not Discussed:**

The method focuses on the interpretability of VQ-tokenizer tokens, but the paper lacks analysis and discussion of other state-of-the-art tokenizers. For instance, it does not compare or analyze methods such as Residual Quantization, VAR, and MAGVIT2, which are considered excellent tokenizers. Including such comparisons and discussions would provide a more comprehensive understanding of the context and contributions of the proposed method.

**Experimental Designs Or Analyses:**

Please refer to the section of Methods And Evaluation Criteria

**Methods And Evaluation Criteria:**

The proposed methods and evaluation criteria in the paper generally make sense for the problem or application at hand. The experiments are primarily conducted on VQGAN models trained on the ImageNet dataset to verify the relationship between tokens and categories. However, the benchmark used for validating the token and category relationship is not particularly authoritative, and there is a notable lack of comparison with other existing methods.

**Other Comments Or Suggestions:**

Considering these issues, my inclination is to give a weak reject. I believe the authors need to address these questions. Of course, if they can provide convincing answers, I would consider raising my score.

**Other Strengths And Weaknesses:**

This paper primarily focuses on the interpretability analysis of tokens and their corresponding semantic categories within image tokenizers. The methodology involves using a pretrained generative model (e.g., VQGAN) to generate samples, then training an IEM based on these generated samples and their corresponding categories. The IEM is subsequently used to calculate the relationship between tokens and categories in the image. Frankly, I find this framework quite interesting, but I have the following concerns:

1. I believe the interpretability of tokens offers limited technical contribution to the enhancement of generative model performance and the optimization of future tokenizers.
2. I feel that the benchmark constructed in this paper is not very clear. How to evaluate the effectiveness of this work on the benchmark itself is not clearly explained.
3. This work lacks comparisons with other image tokenizers. It is uncertain whether different tokenizers will exhibit various forms of performance.
4. Additionally, there are no experimental validations on T2I models. If this experiment were conducted on T2I models, considering the absence of category IDs, how would the association between tokens and text be calculated?
5. Additionally, I found that the interpretability methods themselves lack significant innovation; these methods are mostly an application of previously established interpretability techniques from the visual recognition field (refer to CAM or Grad-CAM) applied to tokenizer recognition.

**Questions For Authors:**

please refer to "Other Strengths And Weaknesses"

**Relation To Broader Scientific Literature:**

This paper focuses on interpretability-related work and its broader research value is somewhat limited.

**Theoretical Claims:**

I have a good understanding of the derivations presented in the Sample-level Explanation and Codebook-level Explanation sections. The mathematical formulations and explanations provided in these two sections are relatively clear and well-structured, making it easier to follow the authors' reasoning and methodologies.

---

> ### Author Rebuttal · Authors · 2025-04-01
>
> Thank you for your thoughtful review. We address your concerns as follows.
>
> **Limited technical contribution to VQGMs:**
> Our method provides a **quantifiable** approach to **detecting bias** in VQGMs and pinpointing the **specific tokens responsible**, enabling targeted debiasing and image editing. This interpretability can be used to improve fairness, which is a key aspect of generative model performance, by facilitating both the detection of bias and the identification of its underlying sources.
>
> **Unclear benchmark evaluation:**
> Here, we construct a comprehensive evaluation protocol to measure whether CORTEX-selected tokens are truly relevant to the target concept. It is worth noting that no off-the-shelf benchmark exists for evaluating token-level interpretability in VQGMs. Specifically, we conduct three evaluations:
>
> 1. **Sample-level evaluation** (Section 4.2 Figure 4 and Table 2):
>    - **Dataset:** VQGAN-generated ImageNet (comprehensive visual concepts coverage).
>    - **Evaluation Metric:** Drop in pretrained classifier accuracy (ViT/ResNet) after masking identified tokens; greater drops indicate higher importance of selected tokens.
>    - **Results:** Masking CORTEX-identified tokens causes significantly larger accuracy drops compared to randomly or frequency-based selected tokens, confirming that CORTEX effectively identifies concept-critical tokens.
>
> 2. **Codebook-level evaluation** (Appendix A.4 Table 5):
>    - **Dataset:** VQGAN-generated 10 bird categories (chosen because pretrained classifiers achieve high accuracy on these categories, ensuring reliable evaluation metrics).
>    - **Evaluation Metric:** Increase in target label probability ($\Delta P_{\text{Targ}}$) when selecting small token regions; larger $\Delta P_{\text{Targ}}$ indicates stronger association between selected tokens and target concept.
>    - **Results:** CORTEX consistently achieves higher $\Delta P_{\text{Targ}}$ values compared to baseline methods, indicating that CORTEX selects tokens that are more aligned with the target concept.
>
> 3. **DALLE-mini bias detection** (Section 5.1 Table 4):
>    - **Dataset:** Images generated by DALLE-mini using neutral prompts ("a doctor in the hospital").
>    - **Evaluation Metric:** Frequency comparison of tokens associated with "white doctor" vs. "black doctor."
>    - **Results:** Tokens representing white doctors occur four times more frequently than those representing black doctors, demonstrating clear bias in DALLE-mini’s generated outputs.
>
> We will provide a more detailed description of our benchmark and evaluation in the revision.
>
> **Comparisons with other tokenizers:**
> Our proposed CORTEX can be applied to any vector-quantized generative model. We chose VQGAN as our backbone since it is a highly representative method in the field.
>
> Also, following your suggestions, we further validate CORTEX on VAR [1], a SOTA VQGM. Specifically, we trained an Information Extractor Model (IEM) using token-based embeddings generated by VAR. We mask the Top-10, 20, and 30 tokens selected by CORTEX and compare the drop in prediction probability against randomly selected tokens. Evaluation using 10,000 VAR-generated images confirms that CORTEX effectively identifies concept-critical tokens.
>
> | Pretrained Model | Top-10 (ours) | Top-10 (random) | Top-20 (ours) | Top-20 (random) | Top-30 (ours) | Top-30 (random) |
> | ---------------- | ------------- | --------------- | ------------- | --------------- | ------------- | --------------- |
> | ViT-B/32         | **3.9**       | 1.3             | **9.4**       | 3.2             | **15.8**      | 6.7             |
> | ResNet50         | **11.8**      | 4.5             | **31.3**      | 22.5            | **49.4**      | 46.5            |
>
> *Table: Prediction probability drop after masking tokens from CORTEX on VAR-generated images; Larger drops indicate that the masked tokens are more important.*
>
> **T2I model application:**
> We would like to clarify that our method has been applied to a T2I model, as shown in Figure 1 and Table 4 using DALL·E.
> Instead of using category IDs, we directly use the text prompt (e.g., “a black doctor”) as the concept. CORTEX identifies visual tokens that are strongly associated with concept words in the prompt, allowing us to analyze token-concept relationships without requiring predefined labels.
>
> **Innovation in interpretability methods:**
> First, traditional methods like CAM or Grad-CAM interpret models in the pixel space, while our method operates in **discrete token space**.
> Second, traditional methods aim to explain the classifier itself, whereas we leverage the Information Bottleneck principle to train a **classification model that interprets large generative models**.
>
> [1] Tian, et al. "Visual autoregressive modeling: Scalable image generation via next-scale prediction." NeurIPS, 2024.

---

> > ### Comment · Reviewer_1ugt · 2025-04-09
> >
> > Thank you very much for the author's detailed reply. After reading the rebuttal, I feel that this article has certain significance. However, somehow it lacks a bit of technical contribution. Regarding my acceptance opinion of this paper, I tend to think it is borderline, leaning towards acceptance.

---

> > > ### Author Response · Authors · 2025-04-09
> > >
> > > Thank you for your thoughtful comments and for acknowledging the significance of our work. We appreciate your engagement and the consideration that the paper is within reach of acceptance.
> > >
> > > We respectfully ask if you would consider updating your score. Thank you again for your time and consideration.

---

### Decision · Program_Chairs · 2025-05-01

**Decision:**

Accept (poster)

**Comment:**

This paper introduces a method for interpreting the tokens generated by VQ models and demonstrates its potential usage across several applications.

Following the rebuttal, two reviewers responded that their main concerns had been addressed, with one increasing their score to a weak accept. Another reviewer, who had initially recommended acceptance, did not participate in the discussion phase. The remaining reviewer, Reviewer 1ugt, acknowledged the paper’s significance and described their stance as “borderline, leaning towards acceptance.” The remaining concern of `Reviewer 1ugt` is the limited technical contribution, such as how interpretability contributes to improving the quality and performance of the VQ model.

Ultimately, none of the four reviewers opposed acceptance, and no major outstanding issues were raised. The Area Chair agrees with the overall consensus and recommends accepting this submission.